# Comprehensive Evaluation of Different TiO_2_-Based Phosphopeptide Enrichment and Fractionation Methods for Phosphoproteomics

**DOI:** 10.3390/cells11132047

**Published:** 2022-06-28

**Authors:** Jiaran Li, Jifeng Wang, Yumeng Yan, Na Li, Xiaoqing Qing, Ailikemu Tuerxun, Xiaojing Guo, Xiulan Chen, Fuquan Yang

**Affiliations:** 1Key Laboratory of Protein and Peptide Pharmaceuticals & Laboratory of Proteomics, Institute of Biophysics, Chinese Academy of Sciences, Beijing 100101, China; 15127805208@163.com (J.L.); wangjifeng@ibp.ac.cn (J.W.); yumengy@dtu.dk (Y.Y.); lina@ibp.ac.cn (N.L.); qing2904495079@163.com (X.Q.); ailikemu.tuerxun16@mails.ucas.ac.cn (A.T.); xiaojingguo@ibp.ac.cn (X.G.); 2College of Life Sciences, University of Chinese Academy of Sciences, Beijing 100149, China; 3Sino-Danish College, University of Chinese Academy of Sciences, Beijing 101408, China; 4Key Laboratory of Plant Resources and Chemistry of Arid Zone, State Key Laboratory Basis of Xinjiang Indigenous Medicinal Plants Resource Utilization, Xinjiang Technical Institute of Physics and Chemistry, Chinese Academy of Sciences, Urumqi 830011, China

**Keywords:** phosphoproteomics, TiO_2_, phosphopeptide enrichment, phosphopeptide fractionation, non-phosphopeptide excluder, deamidation

## Abstract

Protein phosphorylation is an essential post-translational modification that regulates multiple cellular processes. Due to their low stoichiometry and ionization efficiency, it is critical to efficiently enrich phosphopeptides for phosphoproteomics. Several phosphopeptide enrichment methods have been reported; however, few studies have comprehensively compared different TiO_2_-based phosphopeptide enrichment methods using complex proteomic samples. Here, we compared four TiO_2_-based phosphopeptide enrichment methods that used four non-phosphopeptide excluders (glutamic acid, lactic acid, glycolic acid, and DHB). We found that these four TiO_2_-based phosphopeptide enrichment methods had different enrichment specificities and that phosphopeptides enriched by the four methods had different physicochemical characteristics. More importantly, we discovered that phosphopeptides had a higher deamidation ratio than peptides from cell lysate and that phosphopeptides enriched using the glutamic acid method had a higher deamidation ratio than the other three methods. We then compared two phosphopeptide fractionation methods: ammonia- or TEA-based high pH reversed-phase (HpH-RP). We found that fewer phosphopeptides, especially multi-phosphorylated peptides, were identified using the ammonia-based method than using the TEA-based method. Therefore, the TEA-based HpH-RP fractionation method performed better than the ammonia method. In conclusion, we comprehensively evaluated different TiO_2_-based phosphopeptide enrichment and fractionation methods, providing a basis for selecting the proper protocols for comprehensive phosphoproteomics.

## 1. Introduction

Protein phosphorylation is an essential post-translational modification that can regulate almost all aspects of cellular processes, including metabolism, growth, division, differentiation, apoptosis, and signal transduction pathways [1]. Abnormal phosphorylation is the cause or consequence of many diseases [2]. Approximately one-third of all proteins in eukaryotic cells can be phosphorylated at any given time [3]. Phosphorylation primarily occurs on the side-chain hydroxyl groups of serine (Ser, S), threonine (Thr, T), and tyrosine (Tyr, Y), which is called O-phosphorylation. Though phosphorylation can also occur on other amino acid residues, including histidine (His, H), lysine (Lys, K), arginine (Arg, R), aspartic acid (Asp, D), glutamic acid (Glu, E), and cysteine (Cys, C), O-phosphorylation is the primary type of phosphorylation in eukaryotes [4]. Due to their chemical stability in acidic and neutral milieu, analytical methods have primarily been developed for O-phosphorylation, making it the best-studied type of phosphorylation in cell biology and phosphoproteomics [5].

Despite advances in mass spectrometry (MS) technology, many challenges exist in phosphoproteomic analysis. First, phosphoproteins (or phosphopeptides) have very low stoichiometry compared to non-phosphorylated proteins (or peptides), which makes them difficult to identify. Second, phosphopeptides ionize with lower efficiency than non-phosphorylated peptides; therefore, removing non-phosphorylated peptides before MS analysis can boost the MS ion signal of phosphopeptides and increase detection sensitivity [6]. Therefore, the development of effective and specific phosphopeptide enrichment methods is essential for performing comprehensive phosphoproteomic analyses.

In the past two decades, many phosphopeptide enrichment approaches have been described, including antibody-based immunoprecipitation [7], immobilized metal affinity chromatography (IMAC) [8], metal oxide affinity chromatography (MOAC) [9], and sequential elution from IMAC (SIMAC) [10], among others [11,12]. IMAC and MOAC are the most popular techniques for phosphopeptide enrichment since they are simple and effective. However, many molecules, including salts, detergents, and other low-molecular compounds in biological samples, can influence the performance of IMAC [13]. MOAC uses metal oxides to electrostatically interact with phosphopeptides for adsorption and purification, while titanium dioxide (TiO_2_) is the most frequently used chemical species for MOAC [14]. Additionally, TiO_2_-MOAC has higher specificity than IMAC [15].

The enrichment specificity of TiO_2_ is affected by various factors, such as the ratio of peptide-to-TiO_2_ beads [16], the composition of the sample loading buffer, and the washing buffer [13]. The type of non-phosphopeptide excluders in the sample loading buffer significantly affects phosphopeptide enrichment specificity [13]. It has been reported that the inclusion of non-phosphopeptide excluders in the sample loading buffer, such as glycolic acid [13], glutamic acid [17], lactic acid [18], citric acid [19], and 2,5-dihydroxybenzoic acid (DHB) [9,20], can effectively decrease the binding of non-phosphorylated peptides to TiO_2_ without affecting the binding of phosphopeptides. However, conflicting results have been reported for the functions of non-phosphopeptide excluders during phosphopeptide enrichment. Larsen et al. demonstrated that adding DHB to a sample loading buffer can effectively remove acidic non-phosphorylated peptides (non-phosphorylated peptides containing acidic amino acids (D and E)) during TiO_2_ phosphopeptide enrichment for Matrix-assisted Laser Desorption/Ionization (MALDI)-MS analysis [21]. However, this method is not appropriate for liquid chromatography-tandem mass spectrometry (LC-MS/MS) analysis, as DHB is assumed to cause the contamination of both the LC system and the inlet of the mass spectrometer [13,18]. Jensen et al. reported that glycolic acid was an efficient non-phosphopeptide excluder in the TiO_2_ micro-column [13]. However, adding glycolic acid to the sample loading buffer increased the non-specific binding of peptides to TiO_2_ [15,18], and non-specific binding increased as the concentration of glycolic acid in the sample loading buffer increased from 0.25 M to 1 M [15]. These conflicting results make it difficult to identify the best method of enriching phosphopeptides suitable for different biological studies. The aforementioned studies [13,15,18] used simple standard protein or peptide mixtures to compare the effects of non-phosphopeptide excluders on phosphopeptide enrichment with TiO_2_. However, very few studies have thoroughly compared TiO_2_ phosphopeptide enrichment methods with different non-phosphopeptide excluders using complex protein samples. In this study, we comprehensively analyzed the effects of four commonly used non-phosphopeptide excluders in the sample loading buffer, including glutamic acid, lactic acid, glycolic acid, and DHB, on phosphopeptide enrichment with TiO_2_ using 293T cell lysate.

Conventional large-scale phosphoproteomic analysis requires laborious offline peptide fractionation before phosphopeptide enrichment to reduce sample complexity and perform in-depth phosphoproteomic analyses. However, this “fractionation before enrichment” strategy requires a large number of samples (usually 2–3 mg of protein/peptide mixture), which is unsuitable when samples are limited. Additionally, this workflow is time- and labor-intensive. Here, we described a “fractionation after enrichment” strategy, which fractionates purified phosphopeptides with StageTip-based HpH-RP chromatography using Triethylamine (TEA) to achieve in-depth phosphoproteome coverage. This strategy can identify more than 30,000 phosphopeptides corresponding to approximately 20,000 localized phosphosites from a 1 mg 293T peptide mixture. This study will provide a basis for selecting suitable phosphopeptide enrichment methods and provide a more effective fractionation strategy for in-depth phosphoproteomics.

## 2. Experimental Procedures

### 2.1. Cell Culture and Lysis

Human kidney epithelial cell lines 293T were cultured in Dulbecco’s modified Eagle’s medium (DMEM) supplemented with 10% fetal bovine serum, 100 U/mL penicillin, and 100 μg/mL streptomycin at 37 °C and 5% CO_2_. Once grown to 90% confluence, the cell culture media were removed, and the cells were washed three times with ice-cold PBS. The 293T cells were then harvested in the lysis buffer containing 8 M urea and 100 mM Tris-HCl (pH 8.5) supplemented with an EDTA-free complete protease inhibitor cocktail (Roche, Basel, Switzerland) and phosphatase inhibitor cocktail (PhosSTOP, Roche). The cells were then lysed with Precellys Evolution homogenizer (Bertin Technologies, Paris, France). After centrifugation at 20,000× *g* for 20 min at 4 °C, the supernatant was collected, and the protein concentration was determined using a BCA (bicinchoninic acid) protein assay kit (Thermo Fisher Scientific, Waltham, MA, USA).

### 2.2. In-Solution Digestion of Proteins

Proteins were reduced with DTT at a final concentration of 10 mM at 30 °C for 1 h. The resulting free thiols were alkylated with IAM at a final concentration of 40 mM for 45 min at room temperature in the dark. The same amount of DTT was subsequently added to remove excess IAM at 30 °C for 30 min. Proteins were then digested with Lys-C (Wako Pure Chemical Industries, Osaka, Japan) at an enzyme/protein ratio of 1:100 (*w/w*) at 37 °C for 3 h. After dilution with 50 mM Tris-HCl (pH 8.0), samples were digested with sequencing grade modified trypsin (Promega, Madison, WI, USA) at an enzyme/protein ratio of 1:50 (*w/w*) at 37 °C overnight. The enzymatic digestion was stopped with formic acid (FA), and the supernatant was collected after centrifugation at 20,000× *g* for 20 min. Peptides were desalted on HLB cartridges (Waters, Milford, MA, USA) and dried in SpeedVac (LABCONCO, Kansas City, MO, USA). After dissolving the desalted peptides with 0.1% FA, the peptide concentration was determined using a BCA peptide assay kit (Thermo Fisher Scientific). The peptides were then split into different amounts according to the different experiments, dried, and stored at −80 °C for phosphopeptide enrichment. An aliquot of peptide digests from each digestion batch was analyzed with LC-MS/MS to obtain peptide identification data from cell lysate.

### 2.3. Phosphopeptide Enrichment with Different Methods

#### 2.3.1. Enrichment of Phosphopeptides Using TiO_2_ with Glutamic Acid

The phosphopeptides were enriched using TiO_2_ with glutamic acid according to the method previously described [16,17], with some modifications. The peptides were resolubilized in a freshly prepared sample loading buffer containing 65% ACN, 2% trifluoroacetic acid (TFA), and saturated glutamic acid (cat. no. G1251, Sigma Aldrich, Burlington, MA, USA) to a final concentration of 2 μg/μL. TiO_2_ beads (5 μm Titansphere, GL Sciences, Tokyo, Japan) were preconditioned with sample loading buffer for 5 min, and the process was repeated three times. The peptides were then incubated with TiO_2_ beads at a peptides/TiO_2_ ratio of 1:6 (*w/w*) for 15 min at room temperature. After pelleted TiO_2_ beads, the supernatant was transferred to another tube and incubated with half the amount of TiO_2_ beads used in the first incubation. The third incubation was performed with 1/4 the amount of TiO_2_ beads used in the first incubation. TiO_2_ beads from three incubations were pooled with loading buffer and transferred to preconditioned C8 StageTips. The TiO_2_ beads were sequentially washed with sample loading buffer, washing buffer 1 (65% ACN and 0.5% TFA), and washing buffer 2 (65% ACN and 0.1% TFA). The times and/or volume of the loading and washing buffers were the same for all four methods. The phosphopeptides were then eluted with elution buffer 1 (4% ammonia (NH_3_·H_2_O) (Sigma Aldrich)) and elution buffer 2 (4% NH_3_·H_2_O and 50% ACN). The eluted phosphopeptides were immediately acidified with 10% FA and dried in SpeedVac. The phosphopeptides were then desalted with homemade OLIGO^TM^ R3 (Thermo Fisher Scientific) C18 StageTips before LC-MS/MS analysis.

#### 2.3.2. Enrichment of Phosphopeptides Using TiO_2_ with Lactic Acid

The enrichment of phosphopeptides using TiO_2_ with lactic acid was performed according to the method previously described [22], with some modifications. The phosphopeptide enrichment procedure using lactic acid was consistent with the protocol for glutamic acid described above, except that different sample loading and washing buffers were used. The sample loading buffer was 70% ACN, 5% TFA, and 20% lactic acid (cat. no. L6661, Sigma Aldrich); washing buffer 1 was 30% ACN and 0.5% TFA; and washing buffer 2 was 80% ACN and 0.4% TFA.

#### 2.3.3. Enrichment of Phosphopeptides Using TiO_2_ with Glycolic Acid

The enrichment of phosphopeptides using TiO_2_ with glycolic acid was performed according to the method previously described [23], with some modifications. The phosphopeptide enrichment procedure using glycolic acid was consistent with the protocol for glutamic acid described above, except that different loading and washing buffers were used. The sample loading buffer was 80% ACN, 5% TFA, and 1 M glycolic acid (cat. no. 12473-7, Sigma Aldrich); washing buffer 1 was 80% ACN and 1% TFA; and washing buffer 2 was 20% ACN and 0.1% TFA.

In addition to the normal glycolic acid phosphopeptide enrichment protocol, there were other four modified phosphopeptide enrichment protocols that used glycolic acid: (1) PNGase F: the phosphopeptide enrichment procedure was consistent with the normal glycolic acid enrichment protocol, except that peptides were digested with PNGase F (New England BioLabs, Ipswich, MA, USA) at 37 °C overnight to remove glycans from peptides before phosphopeptide enrichment with TiO_2_. (2) 2X washings: the volume of the loading and washing buffers was double that of the normal glycolic acid enrichment protocol. (3) Changed washing buffer: washing buffer 1 was 50% ACN and 0.5% TFA, washing buffer 2 was 50% ACN and 0.1% TFA, and washing buffer 3 (80% ACN and 0.4% TFA) was used. (4) Double TiO_2_: phosphopeptides were enriched with TiO_2_ beads using the normal glycolic acid enrichment protocol described above. After they dried, the enriched phosphopeptides were subjected to a second round of TiO_2_ enrichment using the method previously described [24]. Briefly, phosphopeptides enriched from the first round of TiO_2_ were resolubilized in 70% ACN and 2% TFA and incubated with TiO_2_ beads at a peptide/TiO_2_ ratio of 1:6 (*w/w*) for 15 min at room temperature. After centrifugation, TiO_2_ beads were transferred to preconditioned C8 StageTips, and the supernatant was collected as flow-through (FT). The TiO_2_ beads were then washed with 50% ACN and 0.1% TFA, and the supernatant was collected and pooled with the FT fraction. The pooled supernatant (FT and washing) was designed as TiO_2_-TiO_2_-FT. Next, phosphopeptides were eluted from TiO_2_ beads with the same elution buffers described above.

#### 2.3.4. Enrichment of Phosphopeptides Using TiO_2_ with DHB

The enrichment of phosphopeptides using TiO_2_ with DHB was performed according to the method previously described [25], with some modifications. The phosphopeptide enrichment process using DHB was consistent with the protocol described above, except that different sample loading and washing buffers were used. The sample loading buffer was 80% ACN, 5% TFA, and 20 mg/mL DHB (cat. no. 149357, Sigma Aldrich); washing buffer 1 was 30% ACN and 1% TFA ; washing buffer 2 was 50% ACN and 1% TFA; and washing buffer 3 was 80% ACN and 1% TFA.

### 2.4. StageTip-Based HpH-RP Fractionation of Phosphopeptides

Phosphopeptides were enriched from 1 mg peptides using both the glutamic acid and lactic acid methods. Three biological replicates were performed for each method. After enrichment, 15% of the enriched phosphopeptides were used for single-shot LC-MS/MS analysis, and/or the rest of the samples were used for HpH-RP fractionation.

Two HpH-RP protocols were used to fractionate phosphopeptides using C18 StageTips, and the identification results were compared. C18 StageTip was prepared by plugging a layer of C18 disk (3 M Empore) into a 200 μL pipet tip and methanol-washed C18 beads (cat. no DC930010-L, 3 μm, 150 Å, Agela Technologies, Tianjin, China) were transferred into pipet tips.

For the HpH-RP fractionation of phosphopeptides using ammonia, C18 StageTips were washed with ACN and equilibrated with buffer A (0.1% NH_3_·H_2_O, pH 10) before sample loading. Phosphopeptides were then reconstituted in buffer A and loaded onto StageTips, and the FT fraction was collected. StageTips were then washed with buffer A once, and the washing solution was combined with FT fraction. Then, phosphopeptides were fractionated into five fractions by a stepwise gradient of ACN (2%, 5%, 8%, 15%, and 35%) in 0.1% NH_3_·H_2_O. After elution, the eluates were immediately acidified with 10% FA. Finally, the six fractions were combined into three fractions for LC-MS/MS analysis: 2% ACN/0.1% NH_3_·H_2_O fraction and 15% ACN/0.1% NH_3_·H_2_O fraction were combined as Fraction 1, FT/washing fraction and 5% ACN/0.1% NH_3_·H_2_O fraction were combined as Fraction 2, and 8% ACN/0.1% NH_3_·H_2_O fraction and 35% ACN/0.1% NH_3_·H_2_O fraction were combined as Fraction 3.

For the HpH-RP fractionation of phosphopeptides using TEA, C18 StageTips were washed with ACN and equilibrated with buffer A (0.1% TFA) before sample loading. Then, phosphopeptides were reconstituted in buffer A and loaded onto the StageTips. After washing the tips with H_2_O, phosphopeptides were fractionated into five fractions with a stepwise gradient of ACN (2%, 5%, 8%, 15%, and 35%) in 0.1% TEA (pH 10). After elution, the eluates were immediately acidified with 10% FA. Finally, the five eluates were combined into three fractions: 2% ACN/0.1% TEA fraction and 15% ACN/0.1% TEA fraction were combined as Fraction 1, 5% ACN/0.1% TEA fraction was set as Fraction 2, and 8% ACN/0.1% TEA and 35% ACN/0.1% TEA fraction were combined as Fraction 3.

All samples were dried in SpeedVac and desalted with homemade OLIGO^TM^ R3 C18 StageTips before analyzing with LC-MS/MS.

### 2.5. LC-MS/MS Analysis

All samples were analyzed on an Easy-nLC 1200 HPLC system (Thermo Fisher Scientific) coupled to an Orbitrap Exploris 480 (Thermo Fisher Scientific) with a high-field asymmetric waveform ion mobility spectrometry (FAIMS) device (Thermo Fisher Scientific). All samples were reconstituted in 0.1% FA and separated on a fused silica trap column (100 μm ID × 2 cm) in-house packed with reversed-phase silica (Reprosil-Pur C18 AQ, 5 μm, Dr. Maisch GmbH, Baden-Wuerttemberg, Germany) coupled to an analytical column (75 μm ID × 20 cm) packed with reversed-phase silica (Reprosil-Pur C18 AQ, 3 μm, Dr. Maisch GmbH). The phosphopeptides were analyzed with 132 min gradient (buffer A: 0.1% FA in H_2_O, buffer B: 80% ACN, 0.1% FA in H_2_O) at a flow rate of 300 nL/min (0–5% B, 6 min; 5–20% B, 69 min; 20–30% B, 39 min; 30–99% B, 9 min; 99% B, 9 min). The analysis of 293T peptide digests was performed using 73 min gradient (4–10% B, 3 min; 10–20% B, 22 min; 20–30% B, 20 min; 30–40% B, 15 min; 40–95% B, 3 min; 95% B, 10 min).

MS data were acquired using an Orbitrap mass analyzer in data-dependent acquisition mode. The cycle time was set as 2 s. The spray voltage of the nano-electrospray ion source was 2.0 kV, and with no sheath gas flow, the heated capillary temperature was 320 °C. Full scan MS data were collected at a high resolution of 60,000 (*m/z* 200) from 350 to 1200 *m/z*. The automatic gain control target was 3 × 10^6^, dynamic exclusion was 30 s, and the intensity threshold was 5.0 × 10^4^. The precursor ions were selected from each MS full scan with an isolation width of 1.6 *m/z* for fragmentation with a normalized collision energy of 28%. For phosphopeptide analysis, MS/MS data were acquired at a resolution of 30,000 (*m/z* 200). The automatic gain control target was 1 × 10^5^, the maximum injection time was 54 ms, dynamic exclusion was 30 s, and the intensity threshold was 5.0 × 10^4^. For peptide analysis, MS/MS data were acquired at a resolution of 15,000 (*m/z* 200). The automatic gain control target was 7.5 × 10^4^; the maximum injection time was 22 ms. The compensation voltage of FAIMS was set as −45 V and −65 V.

### 2.6. Data Analysis and Processing

LC-MS/MS raw data were processed with Proteome Discoverer (PD) (version 2.4.1.15) using the Sequest HT search engine for protein identification. To reduce the influence of chimeric spectra, the precursor detector node in PD was added. The database was the UniProt reviewed human protein database (updated April 2021) with 20,386 protein entries and common contaminants. The database searching parameters were set as follows: enzyme specificity for trypsin and up to two mis-cleavages was allowed, the minimum peptide length was 6, and the mass tolerances for precursor and fragment ions were set as 10 ppm and 0.02 Da, respectively. Cysteine carbamidomethylation was set as a fixed modification. Variable modifications were set to methionine oxidation, phosphorylation at serine, threonine, tyrosine, deamidation at asparagine (Asn, N) and glutamine (Gln, Q), and acetylation at the N-terminal of proteins. The false discovery rate (FDR) was calculated using the Percolator algorithm provided by PD. FDR on peptide and protein levels was 1%. PhosphoRS localization probability was set to greater than 0.75 [26]. Only phosphopeptides with fully localized sites were regarded as localized phosphopeptides. If there were multiple phosphosites within one phosphopeptide and at least one phosphosite was ambiguous, this phosphopeptide was not regarded as a localized phosphopeptide. The LFQ intensity of phosphopeptides was extracted from raw data, and a normalized abundance of phosphopeptides was used for principal component analysis (PCA) and correlation analysis. The number of non-redundant localized phosphopeptides and localized phosphosites identified were extracted with an in-house Python script.

The composition of amino acids in peptides and phosphopeptides was calculated with Prot pi_Peptide Tool (https://www.protpi.ch/Calculator/PeptideTool, accessed on 27 September 2021), while the GRAVY (grand average of hydrophobicity) index and theoretical isoelectric point (pI) of peptides and phosphopeptides were calculated with the Sequence Manipulation Suite tool (https://www.bioinformatics.org/sms2/protein_gravy.html, accessed on 20 September 2021) [27].

To analyze the motifs of deamidated (phospho)peptides, sequences of amino acids around deamidation sites (deamidated N or Q) were analyzed with the iceLogo resource (https://iomics.ugent.be/icelogoserver/create, accessed on 16 December 2021) [28]. First, the sequence windows of ±5 amino acids around localized deamidated sites were created. Then, non-redundant sequence windows were submitted for iceLogo analysis, and the precompiled Swiss-Prot human database was used as the background. To analyze the motifs of phosphopeptides, sequence windows of ±7 amino acids around phosphorylated amino acids were created and analyzed as described above. For all analyses, the *p*-value was set as 0.05, and the results were presented as a percentage.

SPSS (version 16.0) was used for statistical calculations, and Origin 8.0 was used to produce figures. The density plots, PCA plots, and correlation plot were prepared with RStudio using in-house scripts.

## 3. Results

### 3.1. The Effect of Peptide Amounts Used for Enrichment on the Number of Phosphopeptides Identified and Phosphopeptide Enrichment Specificity

To identify an optimal starting amount of peptides for phosphopeptide enrichment, we enriched phosphopeptides from different amounts of peptides, including 50, 100, 200, 400, and 1000 μg, using TiO_2_ with lactic acid as a non-phosphopeptide excluder. Two replicates were performed for each condition. The amount of TiO_2_ beads and the volume of sample loading buffer and washing buffers were proportional to the starting amounts of peptides. The enrichment procedure and LC-MS/MS parameters were the same for all the analyses.

The number of phosphopeptides identified and phosphopeptide enrichment specificity (measured by the ratio of phosphopeptides identified to all peptides identified) in each MS analysis, including replicate experiments, are shown in Figure 1 and Appendix A. A remarkable tendency to change was observed. The fewest phosphopeptides were identified in samples enriched from 50 μg of peptides. The reproducibility of the two replicates was poor, which is likely due to variable sample loss during enrichment with lower amounts of peptides in each sample. For phosphopeptide enrichment with 100 μg to 400 μg of peptides, the number of phosphopeptides identified increased as the amount of starting peptides increased. The highest number of phosphopeptides was identified when 400 μg of peptides were used for enrichment. More than 17,000 phosphopeptides were identified in each replicate in a single-shot LC-MS/MS analysis. However, the number of phosphopeptides enriched from 1 mg of peptides was slightly lower than that from 400 μg of peptides, which is likely due to lower enrichment specificity (82%). With an average of 92.3% for the two replicates, the highest enrichment specificity was achieved when 200 μg of peptides were used for enrichment. In this case, on average, more than 16,000 phosphopeptides were identified from the two replicates. The enrichment specificity decreased as the amount of peptides used for enrichment increased once more than 200 μg of peptides were used for enrichment. The enrichment specificity for phosphopeptides enriched from 400 and 1000 μg of peptides was lower than that of 200 μg, likely because the sample complexity increased as the amount of peptides increased, while the non-specific binding of non-phosphorylated peptides to TiO_2_ beads also increased. Decreases in the specificity of phosphopeptide enrichment as the sample amounts increased have been previously reported [24]. In this study, 200 μg of peptides were used to enrich phosphopeptides for single-shot LC-MS/MS analysis since they had the highest enrichment specificity.

### 3.2. Comparison of Four TiO_2_-Based Phosphopeptide Enrichment Methods Using Different Non-Phosphopeptide Excluders

To systematically investigate four TiO_2_-based phosphopeptide enrichment methods, four different non-phosphopeptide excluders were used in the sample loading buffer (glutamic acid, lactic acid, glycolic acid, and DHB) to perform a comparison experiment. A list of solvents, including the loading buffers, washing buffers, and elution buffers used for phosphopeptide enrichment in the four methods, is shown in Table 1. The workflow for comparing four TiO_2_-based phosphopeptide enrichment methods is shown in Figure 2A. The digested peptides were split into 12 aliquots (200 μg peptides each) and subjected to phosphopeptide enrichment using different non-phosphopeptide excluders, producing three independent enrichment results for each method.

The number of phosphopeptides identified and phosphopeptide enrichment specificity for each method, including replicate experiments, are shown in Figure 2B and Appendix A. Different identification profiles were observed for each of the four phosphopeptide enrichment methods. In single-shot LC-MS/MS analysis, the glutamic acid method and the lactic acid method could identify approximately 17,000–19,000 phosphopeptides in each replicate, while the glycolic acid method and the DHB method could only identify approximately 12,000 phosphopeptides. The significant discrepancy in the number of phosphopeptides identified from these four methods is likely due to their different phosphopeptide enrichment specificity. The enrichment specificity of the glutamic acid method and the lactic acid method was high, with respective averages of 86.39% and 88.65%. However, the average enrichment specificity of the glycolic acid method and DHB method were only 52.90% and 41.54%, respectively.

The intensities of phosphopeptides identified from three replicates from each of the four methods were analyzed with Heatmap and PCA. As shown in Appendix A, three replicates of each method were clustered together, while data from different enrichment methods were easily separated. PCA analysis yielded the distinctive clustering of four clusters in which individual methods can be distinguished from the other three methods (Figure 2C). These results indicate that each method reproducibly enriches a specific phosphopeptide population.

We then analyzed the combined data of three replicates for each method and identified 22,916 phosphopeptides corresponding to 12,460 phosphosites with the glutamic acid method. A total of 20,902 phosphopeptides corresponding to 12,546 phosphosites were identified with the lactic acid method. The glycolic acid method identified 16,210 phosphopeptides corresponding to 9747 phosphosites, while the DHB method identified 15,288 phosphopeptides corresponding to 9393 phosphosites. After combining the results of all four methods, 29,354 phosphopeptides corresponding to 15,365 phosphosites were identified (Appendix A). The overlap of phosphopeptides identified in the four methods was 10,411, which only accounted for 35.5% of all phosphopeptides identified (Figure 2D). These results indicate that the phosphopeptides identified by the four methods are strongly complementary, while combining different methods could expand phosphoproteome coverage.

### 3.3. Physicochemical Characteristics of Phosphopeptides Enriched by the Four Methods

To identify why each of the four phosphopeptide enrichment methods identified a different population of phosphopeptides, we compared the physicochemical characteristics of peptides identified from cell lysate (the starting material) (Appendix A), and phosphopeptides enriched from the four methods and non-phosphopeptides identified in the four methods (peptides identified in each method that are not phosphopeptides).

First, we investigated the number of phosphosite(s) per phosphopeptide. As shown in Figure 3A, most phosphopeptides were singly-phosphorylated. However, the percentage of multi-phosphorylated peptides (doubly-phosphorylated and triply-phosphorylated peptides) enriched from the lactic acid method was much higher. Approximately 28% of phosphopeptides were multi-phosphorylated in the lactic acid method compared to 18% in the other three methods. A high percentage of multi-phosphorylated peptides enriched by the lactic acid method was previously observed [22]. Therefore, one advantage of the lactic acid method is that it can enrich a higher percentage of multi-phosphorylated peptides.

One challenge for phosphopeptide analysis is unambiguously localizing phosphosite(s) within phosphopeptides, which is important for understanding the roles of phosphorylation events [29]. We analyzed the phosphosite localization rate of phosphopeptides, which is defined as the percentage of phosphopeptides with fully-localized phosphosites. The phosphopeptide localization rate of the lactic acid method was approximately 78% compared to 81%, 81%, and 83% in the glutamic acid method, glycolic acid method, and DHB method, respectively (Figure 3B). The lower phosphosite localization rate could be because a higher percentage of multi-phosphorylated peptides were identified in the lactic acid method, and it is more difficult to localize multi-phosphosites within one phosphopeptide. However, there was no significant difference in the frequency of localized phosphoserine (pS), phosphothreonine (pT), and phosphotyrosine (pY) residues in the four methods (data not shown).

Third, we investigated the percentage of mis-cleavage in phosphopeptides. Approximately 40% of phosphopeptides identified carried at least one mis-cleavage, compared to about 10% in peptides of cell lysate (Appendix A). The high percentage of mis-cleavage in phosphopeptides was previously reported [25,30]. The presence of the phosphoryl group affects the trypsin digestion of proteins and can alter the charge distribution of peptides. The potential electrostatic interaction between arginine/lysine and phosphoamino acids in the phosphorylated sequence impairs the accessibility of the trypsin cleavage sites [30]. However, there was no significant difference in the percentage of mis-cleavage in phosphopeptides identified in the four methods.

Then, we conducted an iceLogo motif analysis of the phosphopeptides identified in each method. In phosphopeptides identified by each method, both pS and pT had a significant bias for proline (P) at the +1 position and bias for acidic amino acids (D/E) and P at the +2 and +3 positions (Figure 3C). We then analyzed the amino acid composition of the phosphopeptides identified by each method and peptides identified from cell lysate and observed a significant bias toward S and P and a slight bias toward acidic amino acids D/E in phosphopeptides (Figure 3D). The bias toward these four amino acids in phosphopeptides enriched with TiO_2_ was previously reported [31]. The amino acid composition result was consistent with the results of the motif analysis. The special amino acid composition and motif of phosphopeptides would significantly impact the deamidation of phosphopeptides, as discussed below. We then analyzed the amino acid composition of non-phosphopeptides and found that non-phosphopeptides displayed a significant bias toward D/E (Appendix A), suggesting that in phosphopeptide enrichment methods using TiO_2,_ the acidic non-phosphorylated peptides bound to TiO_2_ beads, decreasing enrichment specificity.

Lastly, we compared the peptide length, hydrophobicity, and pI of phosphopeptides enriched by the four methods and peptides identified from cell lysate. As shown in Figure 3E, the phosphopeptides were much longer than the peptides, likely due to a high percentage of mis-cleavage in phosphopeptides. For the four methods, phosphopeptides identified in the glutamic acid method were shorter, while the lactic acid method identified longer phosphopeptides. When analyzing phosphopeptides exclusively identified using each method, the average length of phosphopeptides using the glutamic acid method was 20 amino acids, and the average length of phosphopeptides using the lactic acid, glycolic acid, and DHB method was 24, 26, and 24, respectively (Appendix A).

When comparing the hydrophobicity of peptides and phosphopeptides using the GRAVY index, we found that phosphopeptides were much more hydrophilic than peptides identified from cell lysate. Furthermore, phosphopeptides identified by the lactic acid method were slightly more hydrophilic than those of the other three methods (Figure 3F), likely because more multi-phosphorylated peptides were identified by this method. The results were more obvious when analyzing phosphopeptides exclusively identified by each method compared with all phosphopeptides identified by each method (Appendix A).

The theoretical pI values of phosphopeptides identified using the four methods and peptides from cell lysate showed different distributions: peptides had higher pI values than phosphopeptides. In addition, phosphopeptides identified with the lactic acid method were more acidic than from the other three methods (Figure 3G). When analyzing phosphopeptides exclusively identified by each method, we found that the lactic acid method showed a distinct advantage for phosphopeptides with pI values <5.0 (highly acidic) compared to the other three methods, while the glutamic acid method identified more phosphopeptides with pI values >5.0 (Appendix A).

In summary, phosphopeptide enrichment methods using different non-phosphopeptide excluders can purify phosphopeptides with different physicochemical characteristics, such as the number of phosphosites within phosphopeptides, the peptide backbone bearing phosphorylation (measured by peptide length), the hydrophobicity of phosphopeptides (measured by GRAVY index), and the acidity of the peptide backbone (measured by pI). However, we also identified similar phosphopeptide characteristics in the four methods, such as consensus patterns in peptide sequences and similar amino acid compositions.

### 3.4. Improved Phosphopeptide Enrichment with TiO_2_ Using Glycolic Acid

The phosphopeptide enrichment specificity of the glycolic acid method was quite low (about 53%). Since many factors can influence phosphopeptide enrichment specificity, we modified four protocols of the normal glycolic acid method to see if they improved the enrichment specificity. We first considered whether the high selectivity of TiO_2_ toward N-linked sialylated glycopeptides (except for phosphopeptides [32,33]) was the reason for the low enrichment specificity using glycolic acid. Therefore, we removed glycans from sialylated glycopeptides using PNGase F before TiO_2_ enrichment and investigated whether the deglycosylation of peptides could improve phosphopeptide enrichment specificity. Second, we increased the volume of the washing buffers to remove as many non-phosphorylated peptides from TiO_2_ beads as possible. The volume of the loading and washing buffers doubled that of the normal phosphopeptide enrichment protocol (2X washings). Third, we changed the washing buffers to test whether different washing buffers can improve phosphopeptide enrichment specificity. Lastly, a second round of enrichment with TiO_2_ was performed for phosphopeptides enriched from the first round of TiO_2_ enrichment. For each protocol, three replicates were performed.

The number of phosphopeptides identified and the phosphopeptide enrichment specificity of each protocol, including replicate experiments, are shown in Figure 4A and Appendix A; significant changes were observed. Compared with the normal glycolic acid method, phosphopeptide enrichment specificity slightly increased after deglycosylation with PNGase F, and the average phosphopeptide enrichment specificity increased from 52.90% to 57.46%. Increasing the volume of loading and washing buffers increased the phosphopeptide enrichment specificity to 63.28%, while changing the washing buffers increased the phosphopeptide enrichment specificity to 72.48%. However, double TiO_2_ enrichment produced the greatest improvement. On average, the phosphopeptide enrichment specificity of the three replicates increased to 87.42%, which is comparable to that of the glutamic acid method and the lactic acid method described above. The number of phosphopeptides identified increased as the phosphopeptide enrichment specificity increased. There were approximately 13,000 phosphopeptides identified with the normal method and about 18,000 with the double TiO_2_ method. These results indicate that the phosphopeptide enrichment protocol using TiO_2_ with glycolic acid as a non-phosphopeptide excluder is suitable for the enrichment of phosphopeptides with two rounds of TiO_2_. Examples of phosphopeptide enrichment methods using two rounds of TiO_2_ enrichment include the TiSH method [23] and the simultaneous enrichment of phosphopeptides and N-linked sialylated glycopeptides [34].

We also analyzed TiO_2_-TiO_2_-FT, which is the supernatant (FT and washing) of the double TiO_2_ enrichment. Approximately 10% of the peptides identified in TiO_2_-TiO_2_-FT were phosphopeptides, this indicates that these phosphopeptides enriched in the first TiO_2_ enrichment were lost during the second TiO_2_ enrichment. Respectively, 2228, 2240, and 3195 phosphopeptides were identified in the three replicates of TiO_2_-TiO_2_-FT, and there was a total of 3704 phosphopeptides identified in the three replicates of TiO_2_-TiO_2_-FT (Appendix A). However, there was a significant overlap of phosphopeptides between double TiO_2_ (TiO_2_-TiO_2_) and TiO_2_-TiO_2_-FT; approximately 85% of the phosphopeptides identified in TiO_2_-TiO_2_-FT were also identified in TiO_2_-TiO_2_ (Figure 4B). We then investigated why these phosphopeptides were lost during the second TiO_2_ enrichment by comparing the physicochemical characteristics of phosphopeptides identified in both TiO_2_-TiO_2_ and TiO_2_-TiO_2_-FT with phosphopeptides exclusively identified in TiO_2_-TiO_2_-FT. We found that the phosphopeptides exclusively identified in TiO_2_-TiO_2_-FT were longer (Appendix A) and more hydrophobic (Appendix A) than the phosphopeptides identified in both parts. This suggests that longer phosphopeptides and hydrophobic phosphopeptides have lower binding affinity to TiO_2_ beads, making it easier to wash them off during the second TiO_2_ enrichment.

Next, we compared the characteristics of phosphopeptides identified in different samples: 97.59% of the phosphopeptides identified in TiO_2_-TiO_2_-FT were singly phosphorylated peptides, while the percentages in the normal glycolic acid enrichment method (single TiO_2_) and double TiO_2_ enrichment (TiO_2_-TiO_2_) were 83.20% and 82.51%, respectively (Figure 4C). These results indicate that the binding affinity of singly phosphorylated peptides to TiO_2_ beads is lower than that of multi-phosphorylated peptides. This result is consistent with a previous observation that multi-phosphorylated peptides had an extremely high binding affinity to TiO_2_ beads [10].

Additionally, the percentages of pT and pY residues were much higher in TiO_2_-TiO_2_-FT. The average percentage of pT in the three replicates of TiO_2_-TiO_2_-FT was 17.13%, compared to 9.85% and 10.86% in single TiO_2_ enrichment and double TiO_2_ enrichment, respectively. The average percentage of pY in the three replicates of TiO_2_-TiO_2_-FT was 0.93%, compared to 0.43% and 0.4% in single TiO_2_ enrichment and double TiO_2_ enrichment, respectively (Figure 4D). These results suggest that the affinity of pT and pY to TiO_2_ beads is much lower than that of pS.

In summary, high phosphopeptide enrichment specificity using TiO_2_ with glycolic acid can be achieved with double TiO_2_ enrichment. However, some phosphopeptides, which have a low affinity to TiO_2_ beads, could be lost during the second TiO_2_ enrichment. This could lead to inaccurate quantification and decreased phosphoproteome coverage.

### 3.5. Deamidation in Phosphopeptides

Since both the glutamic acid method and the lactic acid method have high enrichment specificity, we enriched phosphopeptides from 1 mg of peptides using both methods. We analyzed enriched phosphopeptides with single-shot LC-MS/MS analysis and found that many phosphopeptides carried at least one deamidation site. An in-depth analysis of the phosphopeptide data of the two methods revealed that the deamidation ratio (measured by the ratio of deamidated (phospho)peptides to all (phospho)peptides identified) was much higher in the glutamic acid method than in the lactic acid method. The average deamidation ratio of the glutamic acid method and the lactic acid method was 21.10% and 13.49%, respectively (Figure 5A and Appendix A). We investigated the deamidation ratio by analyzing the single-shot LC-MS/MS data in all four methods. We observed high deamidation in phosphopeptides enriched by the four methods and a higher deamidation ratio in the glutamic acid method than in the other three methods (Appendix A). These results indicate that a high deamidation ratio is a general phenomenon in phosphopeptides enriched by TiO_2_. The false-positive identification of deamidated peptides could occur in a database search because the ^13^C peaks of amidated peptides can be wrongly assigned as monoisotopic peaks of the corresponding deamidated ones due to a minor difference in mass (19.34 mDa) between them [35]. Deamidated peptides can be reliably identified based on retention time in RP chromatography since deamidated peptides have a different retention time than corresponding amidated peptides. In contrast, amidated peptides and their ^13^C peak have the same retention time [35]. We manually checked MS2 spectra and the retention time of phosphopeptides and their deamidated counterparts. As shown in Appendix A, the diagnostic ions of the deamidation site(s) of phosphopeptides existed in MS2 spectra, and retention time increased as the extent of deamidation increased. These results indicate that deamidation exists in phosphopeptides enriched with TiO_2_.

To identify why the deamidation ratio in phosphopeptides was high, we re-searched cell lysate data by adding deamidation at Asn and Gln as variable modifications. We found that the average deamidation ratio in cell lysate was 4.17%, which is much lower than that of phosphopeptides (Figure 5A). Hao et al. evaluated the deamidation ratio in complex proteomic samples and found that Asn-deamidation occurred in 4–9% of all peptides, and Gln-deamidation occurred at a lower rate (1–4%) [35]. In this study, the overall deamidation ratio in peptides from cell lysate was lower than previously described [35], likely because Tris-HCl buffer (pH8.0) was used as trypsin digestion buffer rather than ammonium bicarbonate, which reportedly produces more Asn-deamidation than Tris-HCl buffer [36]. Though the deamidation of peptides significantly increases under tryptic digestion [35], deamidation in phosphopeptides has not been comprehensively investigated, likely because deamidation was not set as a variable modification in previous phosphoproteomic research.

The sequence motif analysis of residues flanking localized deamidation sites in peptides from cell lysate and phosphopeptides enriched by the glutamic acid method and the lactic acid method had different patterns. As shown in Figure 5B, at the level of cell lysate, the deamidation of Asn was followed by Gly (G), which is consistent with a previous observation that peptides with Asn-Gly sequences had a high degree of deamidation (~70–80%) after standard overnight tryptic digestion (~12 h at 37 °C in ammonium bicarbonate buffer, pH 8.2) [37]. However, there was no specific amino acid following deamidated Gln at the peptide level (Figure 5B), indicating that the deamidation of Gln in peptides was somewhat random during in-solution digestion. For phosphopeptides enriched by both methods, there was a general preference for deamidated Asn and deamidated Gln. Deamidated Asn not only had a significant bias for G at the +1 position, as observed in peptides from cell lysate, but it also had a bias for S at the −5 to +5 position and Pro (P) at other positions (+2 to +5, and −1 to −4). Deamidated Gln had a significant bias for P and S at the −5 to +5 positions. These results are consistent with the observation mentioned above that there is a significant bias toward S and P in phosphopeptides (Figure 3D). A slight bias for acidic amino acids (D/E) was also observed in phosphopeptides enriched using the lactic acid method.

Significant differences in deamidation sequence windows were observed in non-phosphopeptides identified by the glutamic acid and lactic acid methods. For non-phosphopeptides identified in the glutamic acid method, a significant bias for G at the +1 position and acidic amino acids (D/E) at other positions was observed for deamidated Asn. Acidic amino acids (D/E) at the −5 to +5 positions were observed for deamidated Gln (Appendix A). For non-phosphopeptides identified in the lactic acid method, a significant bias for G at the +1 position was observed for deamidated Asn. However, bias toward acidic amino acids in the lactic acid method was not as obvious as in the glutamic acid method. Deamidated Gln had a significant bias for acidic amino acids at the −5 to +5 positions. However, the percentage was much lower than in the glutamic acid method (Appendix A). These results imply that the two enrichment methods have different mechanisms for enriching phosphopeptides.

Next, we investigated deamidation sites in peptides and phosphopeptides. At the cell lysate level, more than 60% of deamidation occurred on Asn (N), while less than 40% occurred on Gln (Q). However, the deamidation of phosphopeptides on Gln increased to approximately 50% in the lactic acid method and approximately 60% in the glutamic acid method (Figure 5C). These results indicate that deamidation on Gln increases in the glutamic acid method.

Lastly, the extent of deamidation was investigated. At the cell lysate level, more than 90% of deamidated peptides had one deamidation site. However, the percentage of phosphopeptides, which had multi-deamidation sites (two or three deamidation sites), was higher than that of deamidated peptides from cell lysate. These results indicate the extent of the deamidation increase in phosphopeptides. The percentage of phosphopeptides with multi-deamidation sites in the glutamic acid method exceeded that of the lactic acid method (Figure 5D).

Altogether, phosphopeptides have a higher ratio of deamidation than peptides from cell lysate. A higher deamidation ratio was observed in phosphopeptides enriched using the glutamic acid method than in those from the lactic acid method. This is a disadvantage of the phosphopeptide enrichment method using glutamic acid, as deamidation can increase the complexities of samples and decrease the identification of non-redundant phosphopeptides.

### 3.6. Comparison of the Ammonia-Based and TEA-Based HpH-RP for the Fractionation of Phosphopeptides

The conventional workflow for the comprehensive phosphoproteomic analysis requires several laborious steps, including the fractionation of peptides into several fractions with strong cation exchange (SCX) chromatography [38] or HpH-RP chromatography [25], after which the phosphopeptides of these fractions are separately enriched. Since the phosphopeptide enrichment specificity of the lactic acid method was high and the deamidation ratio was much lower than the glutamic acid method, we used a workflow combining phosphopeptide enrichment using the lactic acid method and HpH-RP fractionation to perform a comprehensive phosphoproteomic analysis. Phosphopeptides were enriched from 1 mg peptides with the lactic acid method for all analyses. Fifteen percent of the enriched phosphopeptides were used for single-shot analysis, and the rest were used for HpH-RP fractionation. We attempted to fractionate phosphopeptides enriched from 400 μg of peptides into three fractions and found that the number of phosphopeptides identified did not increase much compared with single-shot analysis (data not shown). This is likely because there was a limited number of phosphopeptides enriched from 400 μg of peptides.

Two HpH-RP fractionation methods were applied to fractionate phosphopeptides. In the first, 0.1% NH_3_·H_2_O (pH = 10), which is commonly applied to fractionate peptides in HpH-RP chromatography, was used as an additive to mobile phases of HpH-RP chromatography. In this method, six fractions were collected, including the FT and washing fraction. We identified a significant amount of phosphopeptides, especially multi-phosphorylated peptides, in the FT/washing fraction (data not shown), likely because multi-phosphorylated peptides attain a high number of negative charges at high pH values, reducing the interaction with the RP materials [25]. Six fractions were then combined into three fractions for LC-MS/MS analysis.

The second method used 0.1% TEA (pH = 10) as the additive mobile phase of HpH-RP chromatography. In this method, phosphopeptides were loaded to StageTips in an acidic buffer (0.1% TFA), so there were few phosphopeptides in the FT fraction. After fractionation, five fractions were collected and combined into three fractions for LC-MS/MS analysis. We have also separately analyzed the five fractions from TEA-based HpH-RP and found that expanding three fractions to five fractions increased the phosphoproteome depth by only 15.4% (from 32,578 to 38,487) but required almost twice as much MS time. So, three combined fractions were analyzed in the comprehensive phosphoproteomic analysis.

We compared the performance of the two HpH-RP fractionation methods in phosphopeptide identification. As shown in Figure 6A and Appendix A, the number of phosphopeptides identified in the ammonia-based method was less than that of the TEA-based method. However, the number of phosphopeptides identified in the single-shot analysis was similar, indicating that some phosphopeptides were lost during ammonia-based HpH-RP fractionation. We then investigated the number of phosphopeptides identified from the three fractions of both methods and found that the number of phosphopeptides identified in Fraction 2 of the ammonia-based method was significantly lower than in the TEA-based method. Since Fraction 2 is the combination of FT fraction and 5% ACN-eluate while FT fraction contains many multi-phosphorylated peptides, we investigated the percentage of singly phosphorylated, doubly phosphorylated, and triply phosphorylated peptides in three fractions of both fractionation methods. In Fraction 1, the percentage of triply phosphorylated peptides in the ammonia-based method was significantly lower than in the TEA-based method. In Fractions 2 and 3, the percentage of doubly phosphorylated peptides in the ammonia-based method was significantly lower than in the TEA-based method. However, the percentage of singly phosphorylated peptides was higher in the ammonia-based method than in the TEA-based method (Figure 6B). These results indicate that the ammonia-based method has a disadvantage when identifying multi-phosphorylated peptides (including doubly and triply phosphorylated peptides). Furthermore, the deamidation ratios of each fraction in the ammonia-based and TEA-based methods were not significantly different, likely because the fractionated samples were immediately acidified.

We also investigated the quantitative reproducibility of the three replicates from both fractionation methods. As shown in Figure 6C, the average Pearson correlation between replicates from the same method exceeded 0.95, while the Pearson correlation between different fractionation methods was lower (about 0.89). In addition, the PCA analysis of the three replicates of both methods yielded distinctive clustering in which different methods can be distinguished from each other (Figure 6D). These results suggest that the TEA-based HpH-RP is a better solution for the fractionation of phosphopeptides.

### 3.7. HpH-RP Fractionation of Phosphopeptides to Increase the Depth of Phosphoproteome Analysis

We investigated the phosphopeptide identification results from the TEA-based HpH-RP fractionation, which included the number of peptides, phosphopeptides, phosphosites, and phosphoproteins of the three replicates (Table 2). More than 32,000 phosphopeptides were identified in one replicate in less than 7 h of LC-MS/MS analysis time. The phosphopeptide enrichment specificity after HpH-RP fractionation was slightly lower than in single-shot analysis (82.10% vs. 83.85%). However, HpH-RP fractionation greatly expanded phosphoproteome coverage. The number of phosphopeptides identified after HpH-RP fractionation was nearly twice as high as in the single-shot analysis in each replicate (Figure 7A). The same trend was observed for the number of phosphosites that we identified (Appendix A). The MS intensities of phosphopeptides detected in single-shot LC-MS/MS analysis spanned about five orders of magnitude, while in HpH-RP, it spanned about six orders of magnitude (Appendix A). These results indicate that HpH-RP fractionation increases the depth of the phosphoproteomic analysis and the sensitivity of phosphopeptide detecting.

There was a notable overlap of identified phosphopeptides among the three replicates; approximately 80% of phosphopeptides were identified in at least two replicates (Figure 7B). Based on this strategy, more than 40,000 phosphopeptides corresponding to about 22,000 phosphosites were identified in 293T cells (Figure 7C and Appendix A).

We also assessed the identification information of three fractions in each replicate: on average, 12,554, 13,846, and 11,810 phosphopeptides were identified in Fraction 1, 2, and 3, respectively (Figure 7D). The largest number of phosphopeptides was identified in Fraction 2, likely because of its high enrichment specificity (about 91%). Since phosphopeptides are more hydrophilic than peptides, more non-phosphopeptides were identified in the fraction with a higher percentage of ACN (Fraction 3), which led to lower enrichment specificity (about 70%). The lower enrichment specificity in Fraction 3 reduced the overall enrichment specificity. The separation efficiency of the TEA-based HpH-RP fractionation of phosphopeptides is shown as the percentage of phosphopeptides found in one or more fractions. About 83% of phosphopeptides were identified in only one fraction of each sample (Figure 7E), indicating the good orthogonality of the HpH-RP fractionation system and the following low-pH RP LC-MS/MS system for analyzing phosphopeptides.

In summary, we described a protocol that used TEA-based HpH-RP to fractionate phosphopeptides, which could increase the sensitivity and coverage of phosphoproteome identification.

## 4. Discussion

Due to the relatively low stoichiometry of phosphoproteins in the whole proteome, the specific enrichment of phosphopeptides is essential for the successful analysis of phosphoproteomes. TiO_2_-based MOAC is likely the most common method of enrichment due to its robust protocol, simple procedure, and TiO_2_ selectivity. TiO_2_ enrichment is based on the interaction between the negatively charged phosphate group and the metal oxide. The key to the specific enrichment of phosphopeptides from complex samples with TiO_2_ is to minimize the interference of acidic non-phosphorylated peptides. To improve enrichment specificity, the sample loading buffer for phosphopeptide enrichment should be acidified to pH 2–2.5 with organic acids, such as acetic acid [39] or TFA [20]. The pKa value of phosphate groups (pKa_1_ of phosphoric acid) is 2.15, while the pKa values of the carboxyl groups of aspartic acid and glutamic acid are 3.65 and 4.25, respectively [40]. To reduce the binding of acidic non-phosphorylated peptides to TiO_2_, the pH of the loading buffer should be between the pKa values of acidic amino acids and phosphoric acid. In this way, the negative charge of the carboxyl groups is covered after the protonation of acidic amino acids and no longer binds to positively charged TiO_2_ [12], while most phosphates are in a non-protonated state and still show negative charges. As such, they can bind to TiO_2_.

However, buffer acidification is insufficient for reducing non-specific binding [12]. Non-phosphopeptide excluders are added to the loading buffer as competitors to prevent the adsorption of non-phosphopeptides to TiO_2_ beads. However, the effects of non-phosphopeptide excluders on phosphopeptide enrichment are unclear. This is likely because only simple standard protein or peptide mixtures are used for enrichment, and only MALDI MS, not ESI MS, has been used to acquire MS spectra. In this study, we comprehensively investigated the effects of four commonly-used non-phosphopeptide excluders, including glutamic acid, lactic acid, glycolic acid, and DHB, on phosphopeptide enrichment with TiO_2_ using complex proteomic samples. Phosphopeptide enrichment specificity greatly varied among the four enrichment methods using different non-phosphopeptide excluders (Figure 2B). The selectivity of TiO_2_ in the glutamic acid method was high (>85%), which is consistent with previous observations [17]. The lactic acid method’s high enrichment specificity (about 89%) in this study has also been previously reported [18,22]. In this study, high non-specific binding was observed when 1 M glycolic acid was used as a non-phosphopeptide excluder. These results are consistent with those of Sugiyama et al. [18] but differ from the results from Jensen et al., who found that 1 M glycolic acid could effectively exclude non-phosphopeptides from TiO_2_ micro-columns [13]. An enrichment specificity exceeding 85% can be achieved in the glycolic acid method by washing TiO_2_ beads with ammonium acetate (pH~6.8) to remove non-phosphopeptides from the TiO_2_ beads [41]. However, we did not attempt this method and achieved our greatest improvement in enrichment specificity using the glycolic acid method when double TiO_2_ enrichment was performed (Figure 4A). This indicates that two rounds of TiO_2_ enrichment are required to obtain a high enrichment specificity for the glycolic acid method. However, the double TiO_2_ enrichment method must be approached with caution: some phosphopeptides with a low affinity to TiO_2_ beads could be lost during the second TiO_2_ enrichment (Figure 4B), which would decrease phosphoproteome coverage and compromise quantification accuracy.

DHB was the first non-phosphopeptide excluder used to improve phosphopeptide enrichment specificity in TiO_2_ [9], while phosphopeptide enrichment specificity can reach 90% [25]. However, in this study, the DHB method displayed the lowest enrichment specificity (about 41%). This discrepancy in enrichment specificity could be because the two studies used samples with different complexities. HpH-RP-fractionated or SCX-fractionated peptide mixtures were used in the previous study [25], while an unfractionated peptide mixture from cell lysate was used in this study. Since DHB was assumed to contaminate both the LC system and the mass spectrometer [13], we did not perform further investigations using DHB.

A detailed investigation of phosphopeptide characteristics revealed some differences in the phosphopeptides enriched by the four methods. Compared with the other three methods, the lactic acid method purified more multi-phosphorylated peptides (doubly and triply phosphorylated peptides) (Figure 3A), which led to a lower localization rate of phosphosite(s) within phosphopeptides (Figure 3B). In addition, the difference in peptide length, hydrophobicity, and pI of phosphopeptides were observed for phosphopeptides identified by the glutamic acid method and lactic acid method. For example, shorter phosphopeptides were purified by the glutamic acid method, while more hydrophilic and more acidic phosphopeptides were purified by the lactic acid method (Figure 3E–G). These results indicate that the specificity and selectivity of phosphopeptide enrichment with TiO_2_ depend on the loading conditions with different non-phosphopeptide excluders. Besides that, the phosphopeptide enrichment specificity of TiO_2_ can vary depending on the TiO_2_ beads from different vendors [42]. The extract buffers and enrichment conditions used for phosphopeptide enrichment with different TiO_2_ materials should be optimized to obtain high enrichment specificity [43]. Furthermore, the optimized protocol used here with TiO_2_ may not obtain the best performance for phosphopeptide enrichment with other types of IMAC and MOAC, such as Zr-IMAC, Ti-IMAC, Fe-IMAC, Ga-IMAC, or ZrO_2_, as specific sample-loading conditions should be used for different enrichment materials to increase their performance in phosphopeptide enrichment [44].

Aside from these differences, the phosphopeptides identified by the four methods shared some similarities, such as a high percentage of mis-cleavage (Appendix A), similar peptide sequence patterns (Figure 3C), and a bias towards S, P, D, and E on amino acid composition (Figure 3D). A distinct but partially overlapped population of phosphopeptides was purified by the four TiO_2_-based phosphopeptide enrichment methods. The same results were obtained when comparing phosphopeptides isolated by the three phosphopeptide isolation methods (phosphoramidate chemistry (PAC), IMAC, and TiO_2_) [31]. Since the overlap of phosphopeptides identified by the four methods was only 35.5% of all phosphopeptides identified (Figure 2D), no single method can enrich all phosphoproteome parts, and combined phosphopeptides enriched by different methods could increase phosphoproteome coverage.

Deamidation is a chemical reaction in which an amide functional group on the side chain of amino acids Asn or Gln is removed, typically converting Asn to aspartic acid (Asp) or isoaspartic acid (isoAsp) and Gln to glutamic acid (Glu) or isopyroglutamic acid (γ-Glu) [45]. There are two types of deamidation: enzymatic deamidation, which uses PNGase F to remove N-glycans from N-linked glycoproteins or glycopeptides [46], and nonenzymatic deamidation (also called chemical deamidation), which occurs spontaneously on proteins and peptides both in vivo and in vitro. The latter is discussed in this study.

Previous mechanistic studies have revealed the process of the deamidation of Asn and Gln [47]. Under neutral or alkaline conditions, Asn deamidation proceeds via a succinimide intermediate formed by the nucleophilic attack on the side chain carbonyl carbon of Asn by the backbone nitrogen of the ensuing amino acid residue. The cyclic succinimide intermediate is then rapidly hydrolyzed at either the alpha or beta carbonyl group to produce Asp and isoAsp at a ratio of approximately 1:3 [48]. Under acidic conditions, Asn usually deamidates by direct hydrolysis via acid catalysis [49]. Deamidation on Gln can occur via a mechanism similar to Asn residue by direct hydrolysis at acidic pH or via a glutarimide intermediate at neutral or alkaline pH to yield α-Glu and γ-Glu [50]. In general, the deamidation rate of Gln is much slower than the rate of Asn [45].

The rate of deamidation of Asn and Gln residues in a protein or peptide depends on external conditions, such as buffer type, ionic strength, pH, temperature, and protein or peptide sequence [47]. We speculated that the observed high deamidation ratio in phosphopeptides is likely due to three reasons. First, it has been reported that exposure to elevated pH (>10) increases the rate of the formation of succinimide or glutarimide intermediates due to the greater deprotonation of the peptide bond nitrogen at high pH values. The deamidation reaction proceeded more quickly at high pH, and ammonia was an effective general base catalyst for the deamidation of Asn and Gln residues in peptides [48,51]. The rate of deamidation was 6.5-fold faster in the solution which contained ammonia [48]. Therefore, using a high pH ammonia elution buffer to elute phosphopeptides from TiO_2_ beads would lead to a high deamidation ratio of phosphopeptides. Second, the sequence immediately around Asn and Gln residues significantly affects the deamidation rate. Deamidation proceeds much more quickly if Asn or Gln is followed by a polar amino acid with a relatively small side chain, such as Gly, Ala, Ser, Thr, Asp, Glu, or His; their low steric hindrance leaves the peptide group open for the attack [47]. The rate of deamidation decreases as the size and steric bulk of the residues increase [48]. Neighboring Ser and Thr increase the deamidation rate. In this study, we observed that phosphopeptides enriched by TiO_2_ had a bias toward Ser, Pro, Glu, and Asp at the amino acid level (Figure 3D) and a significant preference for Pro, Asp, Glu, and Ser at the +1 to +3 position of pS and pT at peptide sequence (Figure 3C). From this point of view, the special amino acid sequences in phosphopeptides enriched by TiO_2_ favored the deamidation of Asn and Gln in phosphopeptides. We observed a general preference for amino acids with small side chains in the deamidated phosphopeptides enriched by TiO_2_. There was a high frequency of Gly and Ser at the n + 1 residue of deamidated Asn residue and Pro, Ser, and Glu at the n + 1 residue of deamidated Gln residue (Figure 5B), while a high frequency of Ser, Pro, and Glu before deamidated Asn or Gln was observed (Figure 5B). This indicates that amino acids following Asn or Gln and amino acids before Asn or Gln significantly affect phosphopeptide deamidation. Third, the sample loading buffer for phosphopeptide enrichment with TiO_2_ was acidified with TFA to block the non-specific binding of acidic non-phosphopeptides. The acidic conditions (pH < 2) used during phosphopeptide enrichment could accelerate Asn and Gln deamidation in phosphopeptides by direct hydrolysis.

We then investigated why the deamidation ratio and the percentage of deamidation on Gln were higher in phosphopeptides purified from the glutamic acid method than from the lactic acid method. Since saturated glutamic acid (~0.14 M) was used in the sample loading buffer of the glutamic acid method, we speculate that a high concentration of glutamic acid would lead to the accumulation of glutarimide intermediate, which would accelerate deamidation on Gln in phosphopeptides purified in the glutamic acid method.

Due to the special amino acid sequence and pattern of phosphopeptides and specific conditions for phosphopeptide enrichment, the deamidation of phosphopeptides is a nonnegligible but neglected phenomenon in phosphoproteomic research. The deamidation of phosphopeptides can reduce the intensities of native phosphopeptides and lead to the inaccurate quantification of phosphopeptides, which is especially harmful to low-abundance phosphopeptides.

Due to the complexity of phosphoproteomes, the number of phosphopeptides identified in single-shot analysis is limited, and fractionation of phosphopeptides before LC-MS/MS is essential for increasing phosphoproteome coverage [52]. However, for large-scale phosphoproteomic analysis, it is unclear whether it is better to fractionate peptides first and then perform phosphopeptide enrichment for each fraction (a “fractionation before enrichment” strategy) or to first enrich phosphopeptides from the whole sample and then fractionate purified phosphopeptides (a “fractionation after enrichment” strategy). The first strategy is often used in conventional phosphoproteomic analyses, likely because it has better enrichment specificity. However, this strategy is labor-intensive and time-consuming and requires relatively large amounts of starting material (typically 2–3 mg) since sample loss can occur during the fractionation process. The second strategy is more robust and time-efficient. Hydrophilic interaction liquid chromatography (HILIC) [23] and HpH-RP chromatography [53,54] have been used to fractionate purified phosphopeptides due to their high orthogonality toward acidic RP-LC-MS/MS analysis. In this study, we compared ammonia-based and TEA-based HpH-RP phosphopeptide fractionation methods and found that the TEA-based method performed better than the ammonia-based method, likely because the ammonia-based method loaded samples in high pH conditions, which led to the loss of some multi-phosphorylated peptides since multi-phosphorylated peptides interact less with RP materials under high pH values. With the TEA-based method, on average, more than 30,000 phosphopeptides, corresponding to about 19,000 phosphopeptides, can be identified in three fractions of one replicate experiment (Table 2). Therefore, HpH-RP fractionation could greatly expand phosphoproteome coverage compared with single-shot analysis (Figure 7A).

Altogether, we comprehensively evaluated different phosphopeptide enrichment and fractionation methods, which provides a basis for choosing the best method of performing a comprehensive and in-depth phosphoproteomic analysis. We also provided a robust, efficient, and reproducible large-scale phosphoproteomic analysis workflow, which contributes to a better understanding of phosphorylation-related mechanisms.

## Figures and Tables

**Figure 1 cells-11-02047-f001:**
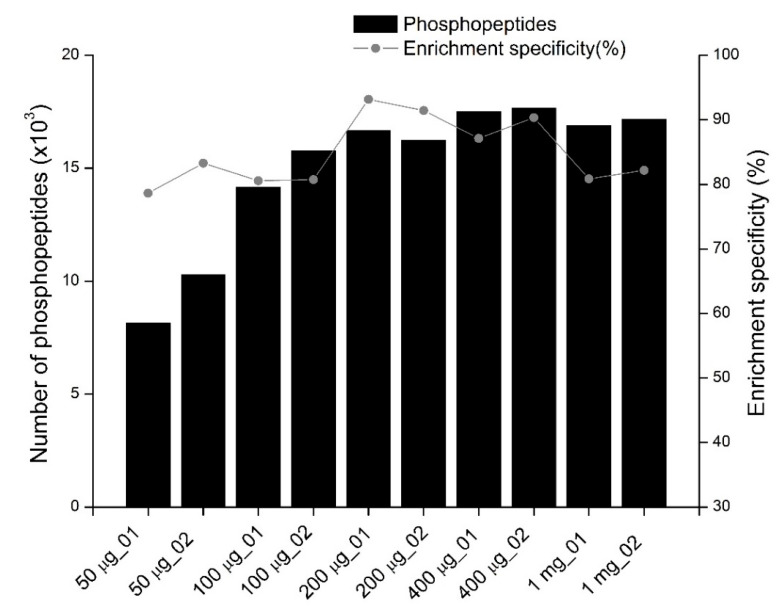
The profiles of phosphopeptides identified from samples in which different amounts of peptides are used for phosphopeptide enrichment. The bar and the curve indicate the number of phosphopeptides identified and the phosphopeptide enrichment specificity (the number of phosphopeptides identified divided by the total peptides identified in each experiment), respectively. The *x*-axis shows the amounts of peptides used for phosphopeptide enrichment. Two replicates are performed for each condition. For all experiments, phosphopeptides are purified with TiO_2_ using lactic acid as a non-phosphopeptide excluder.

**Figure 2 cells-11-02047-f002:**
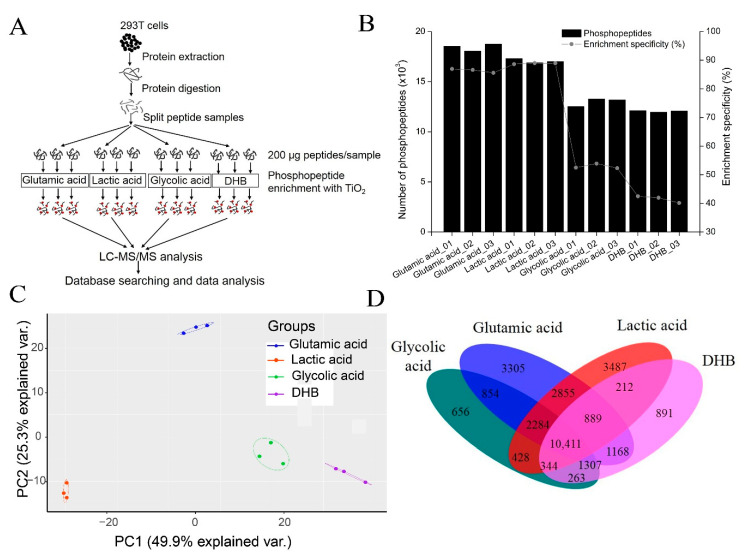
Comparison of four TiO_2_-based phosphopeptide enrichment methods using different non-phosphopeptide excluders. (**A**) The strategy used to compare four different phosphopeptide enrichment methods using TiO_2_ with different non-phosphopeptide excluders, including glutamic acid, lactic acid, glycolic acid, and DHB. (**B**) The profiles of the number of phosphopeptides identified and the phosphopeptide enrichment specificity of each method. Three replicates are performed for each method. (**C**) PCA analysis of the normalized intensities of phosphopeptides identified with the four methods. Each dot represents one replicate of the four methods. Samples from different methods are indicated by color. (**D**) The overlap between phosphopeptides identified with the four methods.

**Figure 3 cells-11-02047-f003:**
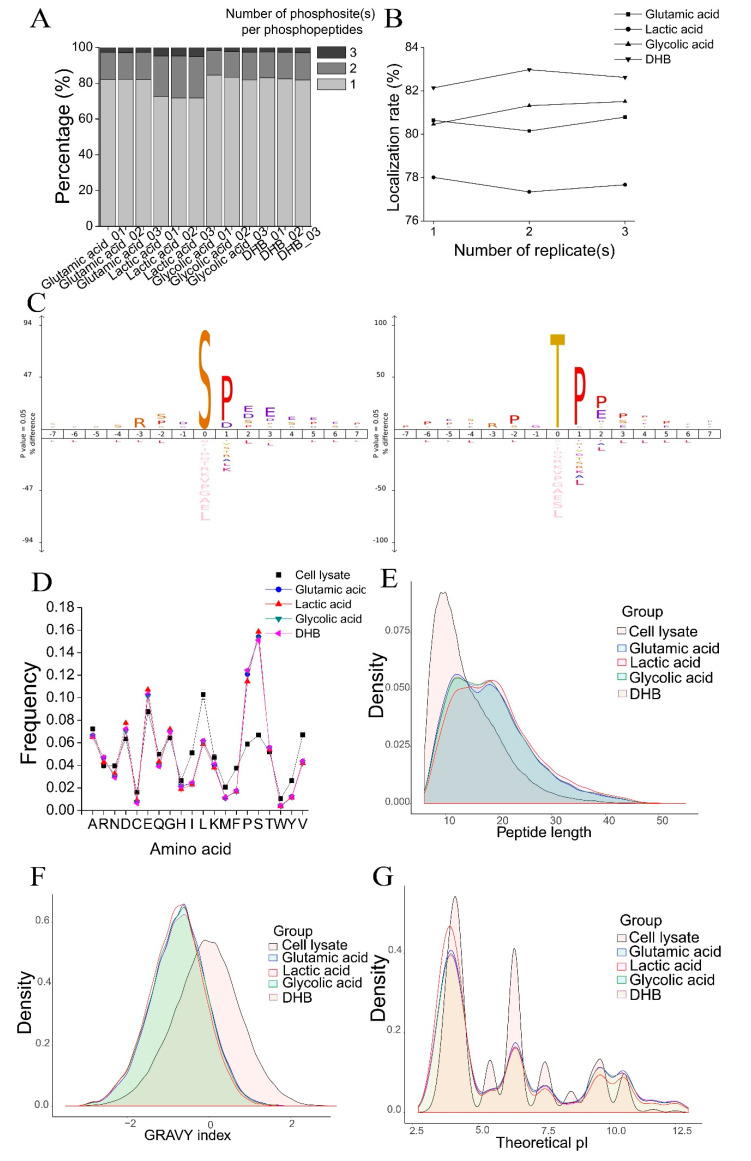
Comparison of the physicochemical characteristics of phosphopeptides identified with the four TiO_2_−based phosphopeptide enrichment methods. (**A**) The percentages of singly phosphorylated, doubly phosphorylated, and triply phosphorylated peptides in the three replicates of the four methods. (**B**) The phosphopeptide localization rate (the percentage of phosphopeptides with fully−localized phosphosites) in the three replicates of the four methods. (**C**) The iceLogo sequence motif analysis of phosphopeptide sequence windows (*p* < 0.05). S represents amino acid serine; T represents amino acid threonine; P represents amino acid proline, D represents amino acid aspartic acid, E represents amino acid glutamic acid. The height of amino acid letters corresponds to the percentage. (**D**) The amino acid composition of peptides from cell lysate (the starting material) and phosphopeptides identified with the four methods. Error bars indicate standard deviation (SD). (**E**–**G**) The distribution of peptide length (**E**), GRAVY index (**F**), and theoretical pI (**G**) of peptides from cell lysate and phosphopeptides identified with the four methods.

**Figure 4 cells-11-02047-f004:**
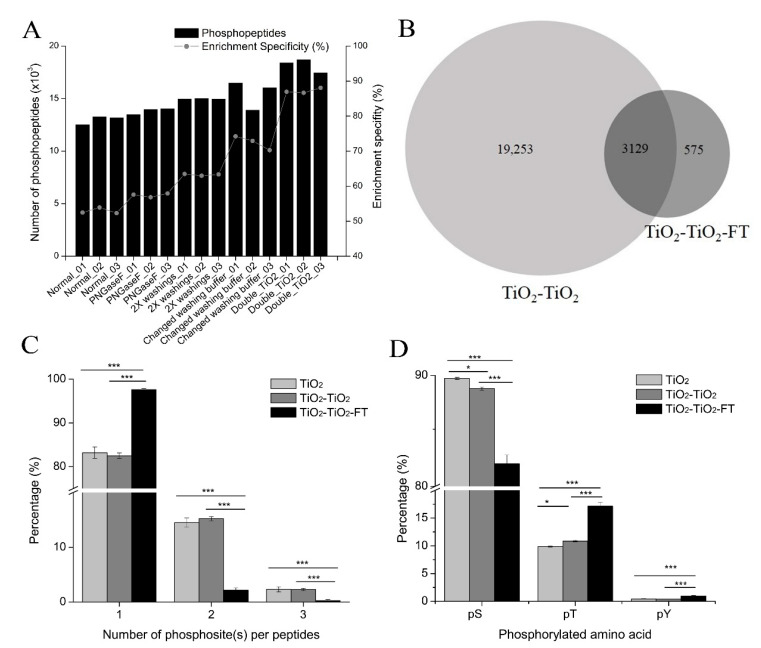
Improved phosphopeptide enrichment protocols with TiO_2_ using glycolic acid. (**A**) Profiles of the number of phosphopeptides identified and phosphopeptide enrichment specificity with the four modified phosphopeptide enrichment protocols that used glycolic acid. Three replicates are performed for each protocol. “Normal” represents the normal phosphopeptide enrichment protocol using glycolic acid; “PNGaseF” represents that PNGaseF is used to remove glycans from peptides before TiO_2_ enrichment; “2X washings” represents that the volume of the loading and washing buffers is double that of the normal phosphopeptide enrichment protocol; “Changed washing buffer” represents that different washing buffers are used during phosphopeptide enrichment; “Double_TiO_2_” represents that two rounds of TiO_2_ enrichment are performed for each experiment. (**B**) Overlap between phosphopeptides identified in TiO_2_-TiO_2_ (double TiO_2_ enrichment) and TiO_2_-TiO_2_-FT (the supernatant of double TiO_2_ enrichment). The combined results of three replicates are shown. (**C**) The percentage of singly phosphorylated, doubly phosphorylated, and triply phosphorylated peptides in TiO_2_ enrichment (normal phosphopeptide enrichment protocol using glycolic acid), TiO_2_-TiO_2_ enrichment, and TiO_2_-TiO_2_-FT. Bars show mean ± SD of three replicates; *** *p* < 0.001 (one-way ANOVA with LSD post hoc test). (**D**) The percentage of phosphorylated amino acids (serine(pS), threonine (pT), and tyrosine (pY)) in TiO_2_ enrichment, TiO_2_-TiO_2_ enrichment, and TiO_2_-TiO_2_-FT. Bars show mean ± SD of the three replicates; * *p* < 0.05, *** *p* < 0.001 (one-way ANOVA with LSD post hoc test).

**Figure 5 cells-11-02047-f005:**
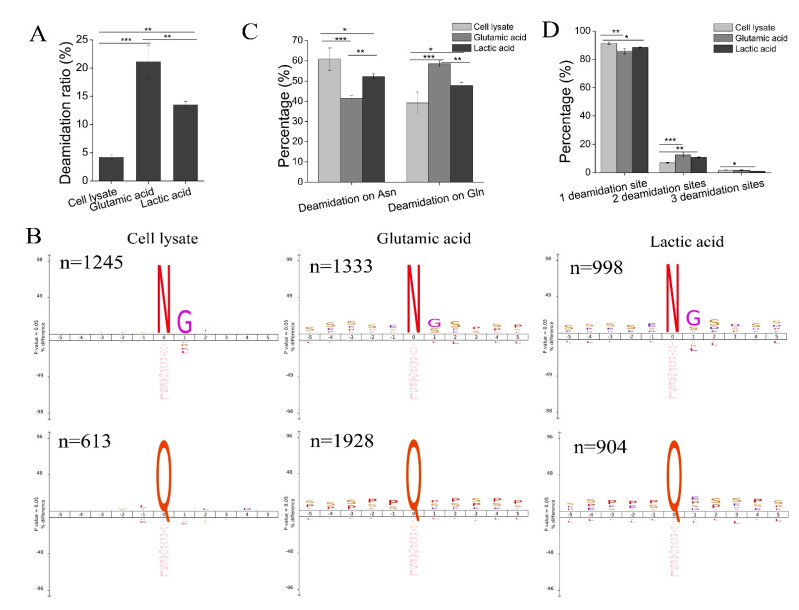
Deamidation of phosphopeptides. (**A**) The deamidation ratio (measured by the ratio of deamidated (phospho)peptides to all (phospho)peptides identified) in peptides identified from cell lysate, and phosphopeptides identified with the glutamic acid method and the lactic acid method. Bars show mean ± SD of the three replicates; ** *p* < 0.01, *** *p* < 0.001 (one-way ANOVA with LSD post hoc test). (**B**) The iceLogo sequence motif analysis of residues flanking deamidation sites in peptides identified from cell lysate (left), and phosphopeptides identified from the glutamic acid method (middle) and the lactic acid method (right) (*p* < 0.05). The height of amino acid letters corresponds to the percentage. N represents amino acid asparagine; Q represents amino acid glutamine; G represents amino acid glycine; S represents amino acid serine. (**C**) The percentage of deamidation sites (deamidation on Asn or Gln) in peptides identified from cell lysate, and phosphopeptides identified from the glutamic acid method and the lactic acid method. Bars show mean ± SD of the three replicates; * *p* < 0.05, ** *p* < 0.01, *** *p* < 0.001 (one-way ANOVA with LSD post hoc test). (**D**) The number of deamidation sites in peptides identified from cell lysate, and phosphopeptides identified with the glutamic acid method and the lactic acid method. Bars show mean ± SD of the three replicates; * *p* < 0.05, ** *p* < 0.01, *** *p* < 0.001 (one-way ANOVA with LSD post hoc test).

**Figure 6 cells-11-02047-f006:**
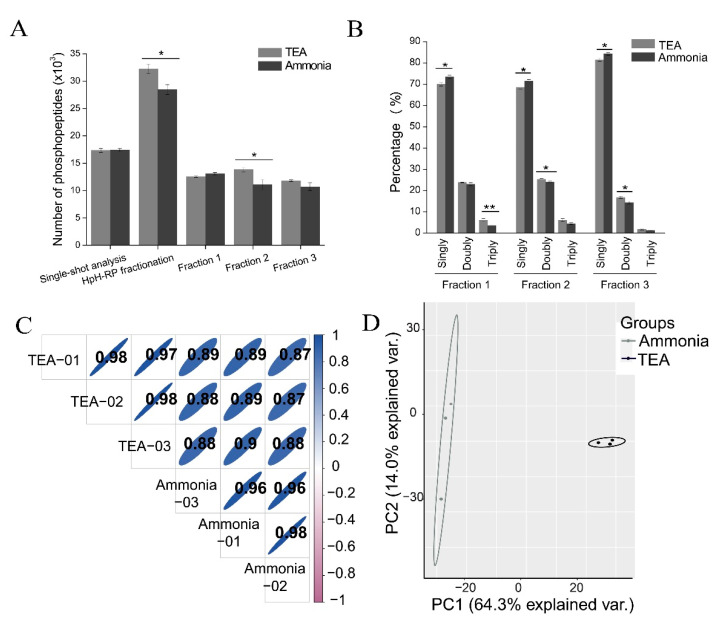
Comparison of the ammonia-based and TEA-based HpH-RP fractionation methods. (**A**) The number of phosphopeptides identified in single-shot analysis (before HpH-RP fractionation), after the ammonia-based and TEA-based HpH-RP fractionation, and three fractions of the ammonia-based and TEA-based HpH-RP fractionation (Fraction 1-3). Bars show mean ±SD of three replicates; * *p* < 0.05 (Student’s *t*-test). For all experiments, phosphopeptides are purified with TiO_2_ using lactic acid as a non-phosphopeptide excluder. (**B**) The percentage of singly phosphorylated, doubly phosphorylated, and triply phosphorylated peptides in the three fractions of the ammonia-based and the TEA-based HpH-RP fractionation. Bars show mean ±SD of three replicates; * *p* < 0.05, ** *p* < 0.01 (Student’s *t*-test). (**C**) Pearson correlation plot of three replicates of the ammonia-based and TEA-based HpH-RP fractionation. (**D**) PCA analysis of three replicates of the ammonia-based and TEA-based HpH-RP fractionation.

**Figure 7 cells-11-02047-f007:**
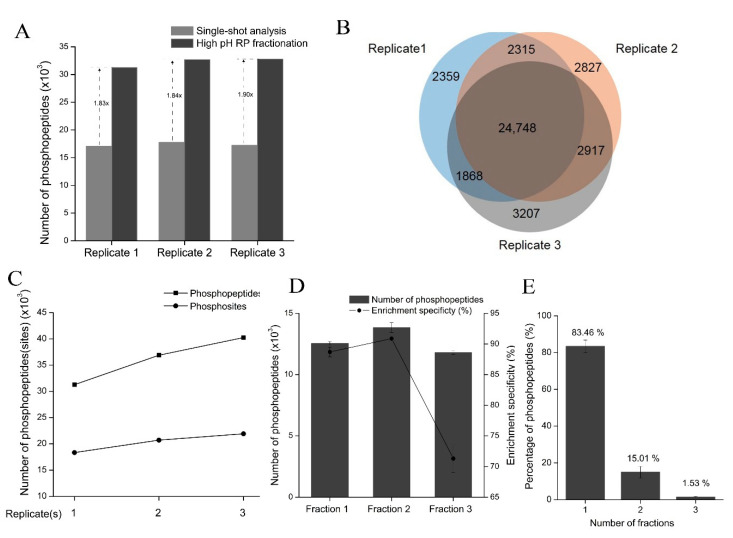
The TEA-based HpH-RP fractionation of phosphopeptides. (**A**) Compared with single-shot LC-MS/MS analysis, HpH-RP fractionation increases the number of phosphopeptides identified from the lactic acid method by nearly two-fold. Three replicates are performed for both single-shot analysis and HpH-RP fractionation. (**B**) The overlap of the phosphopeptides identified in the three replicates of HPH-RP fractionation. (**C**) The cumulative number of phosphopeptides and phosphosites identified in 293T cells. (**D**) The number of phosphopeptides identified and phosphopeptide enrichment specificity in the three fractions of the TEA-based HpH-RP fractionation. Bars show the mean ± SD of three replicates; error bars indicate standard deviation. (**E**) The separation efficiency of the TEA-based HpH-RP fractionation is shown as the percentage of phosphopeptides found in one or more fractions.

**Table 1 cells-11-02047-t001:** List of solvents used for phosphopeptide enrichment in the four methods.

Method	Loading Buffer	Washing Buffer 1	Washing Buffer 2	Washing Buffer 3	Elution Buffer 1	Elution Buffer 2
Glutamic acid	65% ACN, 2% TFA, saturated glutamic acid	65% ACN, 0.5% TFA	65% ACN, 0.1% TFA	____	4% NH_3_·H_2_O	4% NH_3_·H_2_O, 50% ACN
Lactic acid	70% ACN, 5% TFA, 20% Lactic acid	30% ACN, 0.5% TFA	80% ACN, 0.4% TFA	____	4% NH_3_·H_2_O	4% NH_3_·H_2_O, 50% ACN
Glycolic acid	80% ACN, 5% TFA, 1 M Glycolic acid	80% ACN, 1% TFA	20% ACN, 0.1% TFA	____	4% NH_3_·H_2_O	4% NH_3_·H_2_O, 50% ACN
DHB	80% ACN, 5% TFA, 20 mg/ml DHB	30% ACN, 1% TFA	50% ACN, 1% TFA	80% ACN, 1% TFA	4% NH_3_·H_2_O	4% NH_3_·H_2_O, 50% ACN

**Table 2 cells-11-02047-t002:** Identification results of the phosphoproteome of 293T cells using TiO_2_ enrichment with lactic acid and the TEA-based HpH-RP fractionation.

Identification Results	Replicate 1	Replicate 2	Replicate 3
Peptides	37,838	40,431	39,641
Phosphopeptides	31,266	32,730	32,793
Enrichment specificity	82.63%	80.95%	82.72%
Phosphosite	18,322	19,330	19,233
Phosphoproteins	5327	5470	5525

## Data Availability

The mass spectrometry proteomics data have been deposited to the ProteomeXchange Consortium (http://proteomecentral.proteomexchange.org, accessed on 12 May 2022) via the iProX partner repository [55] with the dataset identifier PXD034264.

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
