# Peer review of "Comprehensive Evaluation of Different TiO2-Based Phosphopeptide Enrichment and Fractionation Methods for Phosphoproteomics"

_cells, 2022, doi:10.3390/cells11132047_

Round 1

Reviewer 1 Report

This study describes the effect of different buffers on the efficiency of phosphopeptide enrichment by TiO2. The authors further propose the used of high pH reversed phase fractionation after the enrichment to increase the coverage of phosphopeptides. The approach is certainly not novel, but the results do provide some useful hints for method optimization of phosphopeptide enrichment.

Minor comments:

·   It will be helpful to include a Table with list of solvents (loading buffer, washing buffer, elution buffer etc) and their volume used for phosphopeptide enrichment.

   There are different types of IMAC and MOAC including Zr-IMAC, Ti-IMAC, TiO2, ZrO2, Fe-IMAC, Ga-IMAC etc. Is the optimized protocol in this study applicable to other types of  IMAC/MOAC? The authors should mention this issue in the discussion section.

Reviewer 2 Report

Li et al., compared four TiO2-based phosphopeptide enrichment methods using four different non-phosphopeptide excluders and performed a comparison of two phosphopeptide fractionation methods. The manuscript is well written and there are several useful conclusions that can help in the selection of optimized protocols for comprehensive phosphoproteomics in complex samples. The analysis performed by the authors in very comprehensive, and the data is of very good quality. I would therefore recommend the paper for publication. One point that I would suggest the authors to note is that the conclusions made in this study are based on a particular type of TiO2 beads and it should be discussed whether the conclusions can be generalized when TiO2 beads from a different vendor are used.

Reviewer 3 Report

Although the manuscript does not have a biological hypothesis, I believe that it could be of interest to the scientific community working in the field of proteomics, especially in the field of phosphoproteomics.

The authors provided thea basis for selecting suitable phosphopeptide enrichment methods and provided a more effective fractionation strategy for in-depth phosphoproteomics. This contribution is very important considering that reproducibility of scientific results is one of the most important concerns in science nowadays. I believe that this manuscript, in the field of proteomics, contributes to achieve a methodological standard in this field.

Author Response

Response: Thank you very much for the reviewer’s comments.